# Recent Advances towards the Understanding of Secondary Acute Myeloid Leukemia Progression

**DOI:** 10.3390/life14030309

**Published:** 2024-02-27

**Authors:** Scott Auerbach, Beana Puka, Upendarrao Golla, Ilyas Chachoua

**Affiliations:** 1Department of Molecular Biology and Genetics, Bilkent University, Ankara 06800, Türkiye; scott.auerbach@bilkent.edu.tr (S.A.); beana.puka@bilkent.edu.tr (B.P.); 2Department of Medicine, Division of Hematology and Oncology, Pennsylvania State University College of Medicine, Hershey, PA 17033, USA; ugolla@pennstatehealth.psu.edu

**Keywords:** sAML, AHD, MPN, MDS, tAML, 7 + 3 regimen, AlloSCT, CAR-T, HSC, LSC

## Abstract

Secondary acute myeloid leukemia (sAML) is a heterogeneous malignant hematopoietic disease that arises either from an antecedent hematologic disorder (AHD) including myelodysplastic syndromes (MDS), myeloproliferative neoplasms (MPN), aplastic anemia (AA), or as a result of exposure to genotoxic chemotherapeutic agents or radiotherapy (therapy related AML, tAML). sAML is diagnosed when the number of blasts is ≥20% in the bone marrow or peripheral blood, and it is characterized by poor prognosis, resistance to therapy and low overall survival rate. With the recent advances in next generation sequencing technologies, our understanding of the molecular events associated with sAML evolution has significantly increased and opened new perspectives for the development of novel therapies. The genetic aberrations that are associated with sAML affect genes involved in processes such as splicing, chromatin modification and genome integrity. Moreover, non-coding RNAs’ emerged as an important contributing factor to leukemogenesis. For decades, the standard treatment for secondary AML has been the 7 + 3 regimen of cytarabine and daunorubicin which prolongs survival for several months, but modifications in either dosage or delivery has significantly extended that time. Apart from traditional chemotherapy, hematopoietic stem cell transplantation, CAR-T cell therapy and small molecule inhibitors have also emerged to treat sAML.

## 1. Introduction

Acute myeloid leukemia (AML) is defined as a heterogeneous malignant clonal disorder of hematopoietic stem cells (HSC) and is the most common myeloid disorder among adults. This disease can be secondary (sAML) to either an AHD such as MPN and MDS or as a consequence of a prior treatment (tAML), or without an AHD history in the case of de novo AML [1,2,3]. MPNs lack cytopenia and are instead characterized by heightened differentiation of progenitor cells and are negative for the BCR-ABL fusion protein. They are divided into three main sub-categories: polycythemia vera (PV), essential thrombocythemia (ET), and myelofibrosis (MF) [4]. MPN rates of progression to sAML vary by subtype: on average 15% of PMF patients, 8.35% of PV patients, and 1.85% of ET patients develop sAML over a ten-year period [5]. MDS are narrow clonal stem cell disorders characterized by heightened cytopenia in the blood and bone marrow due to apoptosis of hematopoietic progenitor cells, and one third of these syndromes progress to sAML [6,7]. AA is a rare, life-threatening bone marrow disorder characterized by deficiencies in hematopoietic cell production resulting from T-cell mediated autoimmunity. Like other AHDs, approximately 15–20% of AA patients over a ten-year period progress further to MDS/sAML. In addition to mutations and chromosomal abnormalities (Figure 1), other factors including telomere attrition, time to therapy, and the patient response to initial immunosuppressant treatment contribute to disease progression [8,9].

Unlike AHDs, sAML is a severe disease with a poor prognosis that has an overall survival time of 4.7 months and an event-free survival time of 2.9 months [27]. The disease affects the elderly and the majority of diagnosed cases are over 65 years old: the median age of diagnosis is 67 years old with a third of patients over the age of 75. Common mutations that lead to the evolution of sAML are found in members of the spliceosome such as *SRSF2,* epigenetic modifiers including *TET2*, *IDH1/2*, *ASXL1*, and *EZH2,* or *TP53* which maintains genomic integrity [28]. The aforementioned mutations are acquired on top of the mutations driving MDS or MPN development [29,30,31,32,33,34,35].

According to the 2016 WHO classification, patients are diagnosed with sAML when the percentage of myeloblasts in the bone marrow and/or peripheral blood is equal to or greater than 20% [36,37]. Although blast count has been set as the differentiator between the three phases, there are limitations to this method in that the blasts under examination, commonly through microscopy, are not easily distinguishable between normal samples and those of MDS and MPN patients. In addition to blast count, there are other indicators of progression to sAML, such as decreased apoptosis in the case of post-MDS sAML and increased cell proliferation for post-MPN sAML [38,39,40]. The vast majority of sAML patients progress from MDS (~85%) and upon exposure to therapy (~9%) (Figure 2). MPN and AA contribution is lower with (~5%) and (~1%), respectively [5,7,41,42,43,44].

The outcome of sAML patients correlates with the mutational landscape. For example, patients with *TP53* mutations, roughly 10–15%, have a worse outcome than those with wild type *TP53*. Not just due to the pernicious nature of these mutations, but also because of the co-occurring mutations including but not limited to *IDH2* and *NPM1*. Due to the difference in mutation profiles between sAML and de novo AML, the treatment regimen has not been as successful when applied to sAML. For example, *TP53* mutant patients had an overall survival of 8 months with induction therapy compared with 1.7 months for those without. The hazard ratio for having a *TP53* mutation was 3.1-fold which is higher than either increasing age or performance status, although co-occurring mutations in *FLT3* had a slightly smaller ratio of 3.01 [45]. The most recommended method for treating sAML is allogeneic stem cell transplantation (alloSCT) due to the highest probability of success [46], especially compared with the more traditional method of 7 + 3, or 7 days of continuous dosing of cytarabine followed by 3 days of IV injection of daunorubicin, which has been the standard for decades [47]. However, direct comparison studies have shown that sAML patients are consistently less responsive to 7 + 3 treatment compared with de novo AML, and have a lower overall survival rate with this regimen, prompting the need for new therapies to be developed [48,49]. In this review, we discuss the recent advances in sAML progression, and we elaborate on the factors that drive the clonal evolution and discuss the different approaches used in the treatment of patients.

## 2. Pathophysiology of sAML

The WHO standard of a ≥20% blast count to differentiate between sAML and antecedent disorders is arbitrary like any threshold, but also has a potentially decisive impact on patients who may either display other characteristics of sAML with a blast count below the threshold or do not display characteristics of leukemogenesis despite having surpassed the threshold [50]. Therefore, this approach does not always assure diagnosis accuracy or reflect the complexity of leukemogenesis nor does it guarantee the optimal treatment for the patient. It is crucial to note that there are additional factors such as mutation type influencing diagnosis and treatment regimens. During the progression process, the immune system responds to the growth of the malignant cells. For example, Bauer et al. have shown a shift in immune cell populations between healthy donors’ bone marrow samples and those diagnosed with either MDS or sAML: there were neither CD3+CD8+ nor CD3+FOXP3+ T cells within a 25-micron radius in healthy bone marrow samples, but both of these populations were present in sAML patients at much higher levels. However, when comparing subsets of sAML, the outcomes were not uniform: in contrast to patients with *TP53* mutations, patients with mutations in either signal transduction genes, chromatin modifiers, or splicing factors showed a significant increase in both populations [51].

One of the reasons why sAML is more common in older patients, especially over the age of 60, is the phenomenon of clonal hematopoiesis of indeterminate potential (CHIP), which is defined as having more than 2% mutant hematopoietic stem cells (HSC) without a history of either cytopenia or myeloid neoplasms. The risk for patients accumulates every year by 0.5–1%, thus most sAML patients skew older with some exceptions such as Fanconi anemia (FA) that progresses earlier. FA is a rare blood disorder characterized by a selective growth advantage to HSCs with an extra copy of 1q, leading to bone marrow depletion and an elevated risk of both MDS and later sAML [52,53]. However, in cases of cytopenia where patients do not exactly meet the criteria for MDS, they are diagnosed with clonal cytopenia of undetermined significance (CCUS) [54]. A 2023 study of UK Biobank patients found that patients diagnosed with either CCUS or CHIP had a significantly higher risk to develop MPNs and subsequently sAML: 1.74 for the former and 2.63 for the latter. For comparison, the risk for patients over 65 to develop MPN/sAML was 1.53 and the risk associated with the total number of mutations was 2.32. Non-genomic factors like red blood cell width distribution (RDW) over 15% or the mean corpuscular volume over 100 fL (MCV) had even higher risks: 3.63 and 4.03 [55]. RDW is also a biomarker for leukemic transformation: a higher RDW is not only used to distinguish MDS patients from healthy ones, but it is also a reliable predictor of leukemogenesis years after the initial MDS diagnosis. Higher RDW is associated with overall worse outcome in patients who have been treated with alloSCT and increased possibility to have passenger mutations in genes like NPM1 or ASXL1 [56].

In terms of cytogenetic risk for leukemic transformation, chromosomal mosaicism is also positively correlated with MPN formation: the 10-year cumulative incidence with mosaicism was 83% and 43% without it [55]. Moreover, chromosomal abnormalities impact diagnosis. For example, t(8;21) is associated with a favorable diagnosis, whereas poor prognosis is associated with −7, inv(3)/t(3q)/del(3q), −7/del(7q), or complex karyotype (CK) with ≥3 abnormalities, which substantially increases the risk of leukemic progression [57,58]. A recent study has reported a case of a 44-year-old female with MDS/MPN where constitutional trisomy 21 was the only identified chromosomal abnormality [59]. Due to higher average age, the risk of an adverse karyotype is higher in sAML than in de novo. Compared with de novo AML, sAML patients have lower overall platelet and leukocyte counts, as well as a lower blast percentage in either bone marrow or peripheral blood [60]. Another study found that 81% of post-MDS sAML patients had a lower WBC count compared with 68% of MPN blast phase or 60% of de novo AML cases (Table 1) [61]. 

Another factor that contributes to sAML progression is inflammation through immune system dysregulation. Patients with autoimmune diseases (AIDs) are already at higher risk of developing sAML due to the elevated levels of inflammatory cytokines in the blood [66,67]. A recent study demonstrated that the increase in inflammation is particularly observed in MPN patients with *TP53* mutations, either heterozygous or multi-hit, where the mutant myeloid cells gain a selective advantage over erythroid cells, especially those with WT *TP53*, which leads to a distortion in the ratio between erythroid and myeloid progenitor cells [68]. For example, TNF drives malignant clonal dominance by targeting healthy myeloid progenitor cells with both apoptotic and necroptotic signaling while malignant cells are left unaffected and able to proliferate through immuno-evasive mechanisms. Moreover, constitutive NFκB activity has been reported in both MPN/MDS and sAML patients [68,69,70]. In the case of MDS progressing to sAML, the chemokine receptor CCRL2, normally expressed in granulocytes, monocytes and NK cells is up-regulated in stem cells, which in turn stimulates IL-8 and the chemokine receptor CXCR2 [71,72]. On the other hand, IL-8 is a catalyst for several downstream pathways that promote proliferation, especially in tumors, including *NFκB*, *MAPK*, *AKT*, *STAT3*, and β-catenin. In a study conducted by Montes et al., compared with healthy donors, patients with MDS and sAML have significantly reduced counts of both CD4+ T lymphocytes and NK cells, with sAML having a higher count of CD4+ T lymphocytes than MDS, but lower than healthy donors. This illustrates that at least the correlation between CD4+ T lymphocytes and MDS progression to sAML is not linear. In tandem with lower counts of proactive immune cells, programmed death ligand 1 (PD-L1) is upregulated, which suppresses the T-cell response to tumor growth and permits clonal expansion and metastasis of leukemic cells [73,74]. Patients with sAML have lower or similar expression levels of PD-L1 compared with MPN/MDS, with no difference between early and advanced stages of MDS; suggesting that the peak of PD-L1 expression results in a long-term suppression of the immune response that allows subsequent mutations to develop and trigger the progression to sAML [75]. In parallel, monocytic myeloid-derived suppressor cells (Mo-MDSC), another immuno-suppressive cell type, has been shown to have stronger positive correlation with the progression to sAML from MDS [74,76]. In addition, alteration in the extracellular matrix (ECM), and in particular the leucine-rich proteoglycan biglycan (BG), contributes to the heightened inflammatory environment observed in both MPN/MDS and sAML. BG is expressed in the bone marrow of both MDS and sAML patients but not in healthy individuals: it promotes cell signaling, bone mineralization, and differentiation. The presence of BG was positively correlated with activity of inflammasome components such as IL-1β, IL-18, and IFN-α. There was no significant difference in BG bone marrow expression between MDS and sAML patients. The hazard ratio of BG-high MDS patients versus BG-low patients for progression to sAML was 8.3 [77].

Another key feature of sAML is the increased self-renewal activity of pre- Leukemic Stem Cells (pre- LSC) through the WNT/β-catenin pathway activation during progression, which produces three main LSC phenotypes: multi-potent progenitor (MPP), lymphoid primed multi-potent progenitor (LMPP) and granulocyte-macrophage progenitor (GMP) [78]. Compared with de novo AML, sAML patients had higher amounts of MPP-like LSCs and LMPP-like LSCs, and this difference was more pronounced in post-MPN sAML. Post-MDS sAML patients had more GMP-like LSCs than post-MPN patients, but similar to de novo AML. The first two types of LSCs were strongly correlated with poor prognosis while GMP-like LSCs were more commonly seen in patients (either de novo or sAML) with either an intermediate or favorable prognosis. There was no difference in terms of LSC type distribution between patients younger than 65 and those older than 65 [79].

In contrast to increased pre-LSC activity in MPN patients progressing to sAML, there is a negative correlation between interferon (IFN) activity and risk of sAML. A study by de Castro et al. categorized MPN patients by both LSC and IFN activity and found that those with both the lowest IFN and highest LSC activity had the greatest risk of progression to sAML. Clonogenicity was significantly higher in this cohort compared with the rest of the study population, and the result was the same when comparing before and after transformation. The low IFN activity in these transforming cells also results in a more chaotic microenvironment where endothelial cells are dysregulated, and leukocytes are behaving abnormally while under increased oxidative stress [80].

The transition to sAML is accompanied with a shift in the clonal architecture inside the bone marrow. This shift is correlated with the number of acquired mutations during progression. Static shift occurs when mutations are acquired sequentially and the clones with the most mutations gradually dominate the bone marrow. Dynamic-S (for single nucleotide variant) shift occurs upon acquisition of multiple mutations that can be in multiple categories simultaneously (Table 2) and their rise to clonal dominance is expedited. Finally, the Dynamic-C (for chromosomal) shift is similar to Dynamic-S except that instead of gaining mutations, the clones acquire chromosomal abnormalities that confer a selective advantage (Table 3) [81]. Most genomic aberrations are either initiators in terms of clonal expansion and myeloid transformation or acquired after the process has begun [82].

## 3. Cytogenetics and Mutational Landscape of sAML

Like other cancers, AHD progression to sAML is a gradual process that is accompanied by chromosomal abnormalities and acquisition of mutations in genes involved in several processes such as signaling, splicing, cell cycle, and chromatin modification (Table 2 and Table 3). In MPNs, the most prevalent driver mutation is JAK2 V617F, which accounts for 98% of PV, 55–60% of ET, and MF cases. Frameshift mutations in calreticulin (CALR) represent 20–25% of ET and MF patients. In MDS, the most common mutations affect the members of the spliceosome such as SF3B1, SRDF2, U2AF1, and ZRSR2; and DNA methylation and chromatin remodeling such as TET2, DNMT3A, IDH1/2, and ASXL1 [83]. Before progression to sAML, patients acquire mutations in other genes. Luque Paz et al. performed a molecular study by targeted sequencing on 49 PV and ET patients after leukemic transformation and they found that certain mutations, in particular spliceosome members such as *SF3B1*, were classified as “short-term” mutations that resulted in a rapid transformation, while other mutations, such as *TP53* and *ATM,* were considered “long-term” as they took many years to occur but also had a poor prognosis upon transformation. *TP53* requires both wild-type alleles to be mutated as opposed to other genes and thus it takes more time [84]. Makishima et al. analyzed the mutation landscape in 2250 MDS patients that evolved to sAML and they have shown that the number of mutations, their diversity and clone sizes significantly increased. Based on their mutational landscape, they categorized the patients into two groups. Patients with *FLT3*, *PTPN11*, *WT1*, *IDH1*, *NPM1*, *IDH2*, and *NRAS* mutations were associated with lower risk of progression. Whereas patients with mutations in genes such as *TP53*, *GATA2*, *KRAS*, *RUNX1*, *STAG2*, *ASXL1*, *ZRSR2*, and *TET2* were in the high-risk category [85].

**Table 2 life-14-00309-t002:** Mutated genes implicated in leukemic transformation and clonal expansion.

Category of Genes	Examples	Citations
Spliceosome	*SRSF2, U2AF1, SF3B1*	[86,87]
DNA Methylation	*DMNT3A*, *TET 1/2, IDH 1/2,*	[88,89]
Activated Signaling	*CALR*, *JAK2, PTPN11*, *TpoR*, *KRAS, FLT3, NRAS*	[90,91]
Transcription Factors	*RUNX1, NFE2, TP53*	[92,93,94,95,96]
Chromatin Modification	*EZH2*, *ASXL1, NPM1*	[89,97]

Despite the overlap in the genetic profile, the events accompanying leukemic transformation are not identical between the two disorders: post-MDS sAML is mainly initiated upon acquisition of mutations in: proteins involved in signaling (eg, *K-Ras*, *N-Ras* and *FLT3*), transcription factors (*RUNX1, GATA2, CEBPA*), or nucleophosmin 1 (*NPM1*) [98]. On the other hand, post-MPN sAML is mostly associated with the loss of *TP53*, *RUNX1, IDH1/2, EZH2,* and *ASXL1* (Table 2) [84,99,100]. The rate of leukemic progression varies largely between the three subtypes of MPNs, with MF patients having the highest rate and ET the lowest [5]. A higher proportion of MDS patients progress to sAML: roughly one third will undergo leukemic transformation over a ten-year period [5]. tAML patients usually acquire mutations in genes like *TP53*, *TET2*, *DNMT3A*, *IDH2*, *NRAS*, *RUNX1*, and *SRSF2* before the initiation of therapy. After treatment, they gain mutations in *FLT3*, *IDH1* and *NPM1*, which drive progression [101,102].

Alterations in the TP53 pathway are one of the main drivers of this process; however, the molecular mechanisms behind it are obscure and require further investigations. *TP53* loss by point mutations or chromosomal abnormalities such as gain of 1q that results in the amplification of MDM4, a p53 negative regulator, or deletion of 17p accounts for up to 50% of MPN cases evolving to sAML [19,25,83,92,93,103]. 1q gain is mostly found in PV patients; however, 17p deletions are common in MF patients [25]. *TP53* mutations are also found in post-MDS sAML, but with a lower frequency (5–10%). This frequency increases with age and in patients with complex karyotype or with a loss of chromosomes 5/5q, 7/7q, and 17/17p [104,105,106,107,108]. For tAML patients, *TP53* bi-allelic mutations occur in 25%-50% of the cases making it the most frequently seen mutation in this disease [101,102]. sAML with *TP53* mutations is highly aggressive and characterized by poor prognosis and short overall survival rate [109]. sAML patients usually lose both alleles of *TP53* either by homozygous point mutations, or a point mutation with uniparental disomy (UPD) [19,92,103]. Monoallelic mutations are mostly found in the MDS/MPN-BP (blast phase) stages, and pretreatment stage in tAML; which forms a fertile ground for progression. Upon the loss of the second allele, transformation to sAML is accelerated, indicating a key role for *TP53* in this process.

In a fraction of FA patients evolving to sAML, dysfunction in DNA repair proteins like *BRCA2* as well as duplication 1q have been characterized [53]. Alternatively, other alterations trigger progression to MDS and sAML: either monosomy 7q or duplication of 3q which contains the secondary oncogene *RUNX1.* Both of these chromosomal abnormalities occur after bone marrow cells enter the blast phase [110]. Pezeshki et al. found that roughly 14% of FA patients progressed to sAML, significantly higher than other hematological disorders such as Shwachman–Diamond syndrome or Diamond–Blackfan anemia, and this was caused by the higher abundance of cytogenetic abnormalities [111]. The loss of 7q contributes to sAML because of the triune of LUC7-like proteins, especially LUC7L2, that interact with the spliceosome and regulate exonic splicing. Upon downregulation of *LUC7*-like genes, both intron retention and exon skipping increase. This is clinically relevant given that these genes are disproportionately expressed in the bone marrow and thymus, highlighting their importance [112].

**Table 3 life-14-00309-t003:** Chromosomal abnormalities correlated with sAML transformation.

Type of Chromosomal Abnormality	Examples	Citations
Deletions	del(7q), del(5q), del(17p)	[106,113]
Duplications	dup(1q), dup(3q), dup(11q), dup(17q)	[114,115,116]
Translocations	t(1;11)(q21;p15), t(10;11)(q22;q23), t(8;21)	[65,117,118]
Inversions	inv(3)/t(3;3)	[24]
Monosomy	−7	[87,119]
Trisomy	+8, +19, +21	[59,87,120]
Uniparental disomy	UPD(9p), UPD(1p), UPD (17p)	[75,92,121]

Another important gene that is frequently mutated in sAML is *NPM1*, or nucleophosmin 1, which normally functions as a histone-binding, DNA-stabilizing factor as a response to UV-induced DNA damage, but when overexpressed, it contributes to uncontrolled cell proliferation. *NPM1* is not enough to trigger the progression to sAML on its own, and any mutation in the *NPM1* gene is generally preceded by a mutation in DNA methyltransferase 3A (*DMNT3A*), permitting clonal expansion of hematopoietic stem cells into myeloid progenitor cells. *NPM1* mutant cells are characterized by down-regulation of *TP53* activity and decreased apoptosis as well as *Myc* up-regulation. A third mutation in *FLT3-ITD* (fml-like tyrosine kinase 3) after *NPM1* completes the path towards leukemogenesis, and patients with this combination of mutations, observed in 40% of those who have sAML derived from MDS, have a much poorer prognosis and shorter overall survival rate [2,60,96].

In addition, non-coding RNAs emerge as important players in leukemogenesis. For example, the *miR-320* family of microRNAs (miRNAs), which are down-regulated in many types of cancer, are up-regulated in sAML. Compared with normal patients, both MDS patients with an intermediate-to-high risk for progression and sAML patients had significantly higher levels of these miRNAs expressed in the bone marrow. All *miR-320* family members were negatively correlated with overall survival in MDS patients, but their exact contribution to leukemic progression remains elusive [122]. *miR-196* is involved in MDS progression to sAML by contributing to both increased myeloid cell differentiation and proliferation as well as decreased apoptosis [123]. A miRNA microarray screening revealed a strong correlation between miRNAs associated with cytokine signaling activation, particularly the Toll-like receptor (TLR) family and interleukins, and progression of MDS [124]. Furthermore, the progression is also facilitated by mutations in certain miRNAs. A study performed on 326 patients undergoing alloSCT for sAML after a prior diagnosis of MDS revealed that mutations in *miR-142* are recurrent in these patients [125]. Mutations are not the only source of miRNA alterations: deletions of entire chromosome sections also contribute to their loss of function: the deletion of the chromosome 7q32 coding for pro-apoptotic miRNAs in MDS patients has been strongly correlated with cell proliferation and progression [126]. Additionally, the lncRNA growth arrest-specific transcript 5 (GAS5) acts as both a negative regulator of the oncogenic miR-222 and a positive regulator of the tumor suppressor PTEN. Pavlovic et al. observed lower levels of GAS5 in sAML compared with healthy donors [127].

Extensive sequencing analysis revealed that RNA-binding proteins (RBPs) are essential in normal hematopoiesis, and their mutation is associated with 55% of sAML patients [86,87,128]. RBPs play an important role in RNA splicing, stability, translation, and localization; in addition to controlling the production of different isoforms, which has been reported to impact cancer development through regulation of different mechanisms such as proliferation and differentiation [128]. Moreover, they play key roles in miRNA biosynthesis and maturation. A recent study has reported an inverse correlation between the presence of mature miRNA and the progression to sAML. Bauer et al. observed down-regulation of the ribonucleases Dicer and Drosha in bone marrow samples from sAML patients and attributed heightened immature miRNA levels to the relative sparsity of these proteins [129]. RNA splicing factors *SF3B1* and *SRSF2*, associated with exon skipping and nonsense-mediated decay of homeostatic proteins, contribute to the etiology of sAML and are associated with a poor prognosis in patients. Although elevated levels of mutations in these splicing proteins have been observed in sAML, as well as other myeloid malignancies, the reason as to why these mutations are elevated in these diseases has yet to be fully explained [86]. A major contributor to the pro-inflammatory switch between MPN/MDS and sAML is the change in isoform of adenosine deaminase acting on RNA 1 (ADAR1) from a constitutively active isoform to an isoform selective for inflammatory signaling, thus facilitating leukemogenesis. ADAR1 increases the risk of progression, but it does not act on its own. In tandem with another pro-inflammatory chemokine, apolipoprotein B mRNA editing enzyme catalytic polypeptide like type 3 (APOBEC3), ADAR1 promotes aberrant RNA editing, which enhances alternative splicing of STAT3 into its STAT3β proactive isoform that prevents β-catenin phosphorylation and degradation, allowing the Wnt pathway to continue driving proliferation and leukemogenesis [130].

## 4. Treatment of sAML

The standard of care for sAML patients is the 7 + 3 regimen, which has been improved through a liposomal delivery system CPX-351 (Figure 3) [131]. However, its relative inadequacy towards sAML has propelled a drive towards developing more successful therapeutics. In addition to delivery, this regimen was also modified in an experimental study where cytarabine at a low dose (40 mg/m^2^ for 10 days compared with 100 mg/m^2^ for 7 days for 7 + 3) in combination with the cytotoxic purine analog cladribine. The objective of the low cytarabine study is to test the regimen for patients who were considered unfit for intensive chemotherapy. Overall survival for the entire cohort was seven months, which is promising considering that the average age of the cohort was 70 years old, and 40% of the patients had high risk genetic profiles. However, the survival time for those who had complete remission (CR) was 21 months, also adding to the optimistic prospects for this modified regimen [132]. Taking a more preemptive approach towards treatment, the tumor suppressor FBX011 was identified via CRISPR-Cas9 as a contributor to aberrant RNA splicing via *EZH2* and cytokine-independent growth once it is inhibited. One candidate for preventing the down-regulation of *FBX011* was bortezomib, a proteasome inhibitor already approved by the FDA for the treatment of multiple myeloma and mantle cell lymphoma, but it did not improve the overall survival of sAML patients in a randomized Phase 2 trial. Bortezomib was screened in conjunction with decitabine, a DNA-hypomethylating agent (HMA) currently approved for treatment of MDS. Decitabine has also been used for treating MDS patients that have crossed the 20% blast threshold and progressed towards sAML as an alternative to chemotherapy. HMAs have also been shown to significantly decrease the progression from MPNs to sAML by inducing a viral mimicry response to upregulate IFN activity and reduce LSC levels [80,133,134,135].

Another class of drugs used in combination with decitabine to treat sAML are inhibitors of BCL-2 like proteins, which help LSCs evade apoptotic mechanisms. Venetoclax (BCL-2 inhibitor) has been successfully tested for both de novo AML and tAML in terms of CR, with CR rates for these two AML variants above 70%. However, for post-MDS sAML, the hazard ratio of resistance to treatment is 2.01 compared with de novo AML, and this is likely influenced by two factors: the first is that some of the post-MDS sAML patients had previously received HMA treatment and the second is that some of these patients also had the *RUNX1-RUNX1T1* fusion gene that confers resistance to HMA treatment [136]. Even with the combination of venetoclax and decitabine, it is possible for sAML patients to relapse because of the overexpression of the “don’t eat me” signal CD47. An anti-CD47 antibody, magrolimab, has been developed and is currently being tested on sAML patients in combination with venetoclax and decitabine in ongoing clinical trials [137]. Besides decitabine, another HMA, 5-azacitidine (AZA), has also been screened against post-MDS sAML, especially in patients with *TP53* mutations. AZA and APR-246, a pro-apoptotic agent that restores the normal function of TP53, synergistically suppress AML cell growth by promoting cell cycle stagnation at the G0 phase and apoptosis. APR-246 also halted cell growth in the absence of AZA, but not to the same extent. Another pro-proliferation pathway, FLT3, was also inhibited by this combination. However, the effects of the AZA-APR-246 combination were reversed by the presence of the FLT3 ligand [138].

For post-MPN sAML in particular, there is the promising option of combining inhibitors for both the lysine demethylase LSD1 and the bromodomain and extra-terminal motifs (BET). Using CRISPR knockouts of LSD1 in post-MPN sAML, SET-2 cells demonstrated both increased apoptosis and differentiation compared with control cells. LSD inhibitors used in combination with either ruxolitinib or BET inhibitors do not develop any non-genetic resistance to either of those treatments in sAML xenografts in mice models. Moreover, either of these combinations are proven to be efficient on cells derived from sAML patients suffering from relapse post-3 + 7 treatment [139]. Another recently identified target, that also works synergistically with ruxolitinib and persists in ruxolitinib-resistant post-MPN sAML cells, is CDK9, a transcription-promoting enzyme that helps prolong the lifespan of otherwise short-lived mRNAs for oncogenes like *c-Myc*. When treated with a combination of a CDK9 inhibitor (NVP2) and ruxolitinib, these cells underwent higher levels of apoptosis accompanied with less chromatin accessibility, thus also leading to decreased *Myc* transcription [140,141].

Recent studies present rebecsinib as a novel potential drug that targets LSCs. It specifically inhibits the p150 subunit of ADAR1, which is activated by the inflammatory cytokines and clonal expansion. Another recent comparative whole-genome and whole-transcriptome sequencing analysis of FACS purified pre-LSCs from MPN patients documented APOBEC3C upregulation, increased C to T mutational burden, and HSPC proliferation during evolution. Pre-LSC to LSC evolution is associated with STAT3 editing, STAT3β isoform switching, and increased ADAR1 p150 expression [130]. There was consistently decreased STAT3 phosphorylation and significantly improved survival of treated mice. These studies are notable because they did not just use samples from sAML patients but also from MDS and MPN as well, suggesting that rebecsinib could potentially work as a preventative measure for the progression to sAML and also as a treatment to prevent relapse for sAML patients. ADAR1 is a promising target because of both its strong stimulation of LSC proliferation in an immuno-evasive manner and its weak association with normal myelopoiesis [130,142,143,144].

Another relatively more efficient strategy for sAML treatment is alloSCT (Figure 4), but it has a generally lower response and overall survival rate compared with that of de novo AML. These worse indices are in spite of the higher incidence of graft versus host disease in de novo AML [145]. Nilsson et al. demonstrated the high potential of alloSCT when applied after chemotherapy: the 5-year overall survival rate for sAML patients who had an AHD was 28% for those who underwent transplantation compared with 2% for those who underwent chemotherapy. Post-remission survival rates were significantly higher for sAML patients with alloSCT, but lower than de novo AML (52% versus 65% respectively) [146]. tAML patients, however, since they have a higher risk of relapse because of the presence of comorbidities, show poor outcomes upon alloSCT treatment [21]. A recent study investigated the potential of a CPX-351 combination with alloSCT and showed that 70% of the patients had CR, and 35% of this cohort had *TP53* mutations. Of those with *TP53* mutations, 77% had CR. Cytogenetic risk also did not affect the overall remission rate, and the study group which did not receive alloSCT had a worse performance than those who did [147].

Chimeric Antigen Receptor T (CAR-T) cell therapy has recently emerged as a promising strategy for cancer treatment, since these cells are specifically engineered to bind to biomarkers more commonly expressed on cancer cells [148]. It has been successfully applied in the treatment of sAML in several studies. Zhang et al. targeted the biomarker CLL-1, or C-type lectin-like molecule 1, in one patient with sAML, and the outcome was a morphological, immunophenotypic and molecular CR for over 10 months [149]. It is important to note that the patient in this study was only 10 years old, and age may also be a factor in the future application of this therapy given that the average age of sAML patients is 70 years old. CLL-1, among other several biomarkers including lymphocyte activation molecule CD244 and IL-3 receptor CD123, is disproportionately expressed on LSCs and AML blast cells while simultaneously not detected on normal HSCs. The successes of CAR-T against acute lymphoblastic leukemia and non-Hodgkin lymphoma have also propelled interest in this therapy [146,147,149,150,151].

## 5. Perspectives

Despite the advances and discoveries laid out in this review that provide a greater understanding of the genetic and cytogenetic aberrations associated with sAML progression, more mechanistic studies are needed to uncover the molecular bases behind this process. This would accelerate the development of novel strategies to treat and prevent sAML progression. The successful efforts in the identification of the aberrations in genes involved in this disease should be extended to explore the role of lncRNAs, which have been shown to be involved in cancer etiology and resistance to therapy. Recently, a CRISPR-based study identified the most differentially expressed lncRNAs for AML patients treated with cytarabine through analysis of a corresponding MOLM14 cell line and revealed that lncRNAs associated with oxidative phosphorylation and fatty acid metabolism had the strongest correlation with resistance to treatment and renewed myeloid proliferation [152]. More studies are required to understand their mechanism of action and provide a complete picture of their implication in the leukemogenesis process and resistance to therapy.

CAR-T appears to be a very promising strategy especially after its success in AML. Recently, an improved version of CAR T, called modified or smartly reprogrammed CAR T cells has been developed to overcome toxicity issues in the original strategy [153,154,155,156].

Most recently, at the 2023 ASH conference, several studies introduced novel treatment strategies that would pave the way for better outcomes. Bertulfo et al. showed that a combination of ruxolitinib and CBP30, a bromodomain inhibitor of histone acetyltransferases (HAT) CREBBP and p300, had a synergistic effect on sAML cells viability and resulted in a decrease in leukemia burden [157]. Rahmé et al. demonstrated the potential of alternative intensive chemotherapies (IC) such as CLAG-M (cladribine, cytarabine, G-CSF and mitoxantrone) and FLAG-IDA (fludarabine, cytarabine, G-CSF and idarubicin), or a combination of FLAG-IDA with venetoclax. In a cohort of high-risk sAML patients, FLAG-IDA ± VEN and CLAG-M induced high remission rates. These regimens were associated with limited toxicity and a high rate of transition to alloSCT transplantation (50% of patients), which offered a survival benefit specifically in the FLAG-IDA ± VEN group. As expected, the presence of *TP53* mutation was associated with inferior outcomes [158].

Overall, joint efforts between the experts in the field would help in better understanding the disease, which would have a great impact on the development of novel customized therapeutic approaches.

## Figures and Tables

**Figure 1 life-14-00309-f001:**
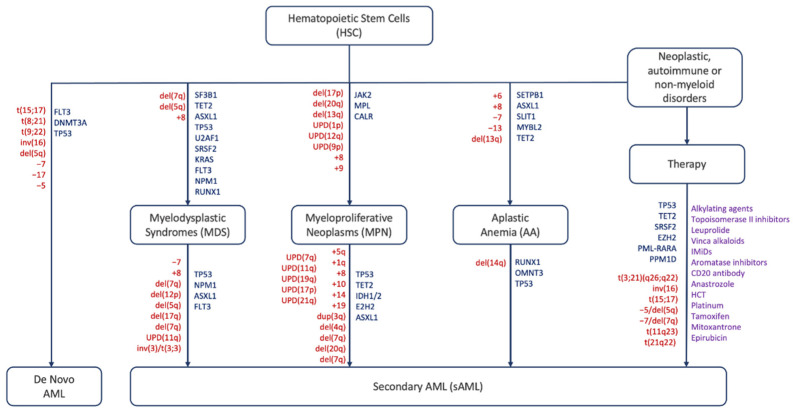
Schematic representation of mutations, abnormalities and factors driving different hematological disorders and their progression to AML/sAML. Red: chromosomal abnormalities, blue: gene mutations, purple: therapeutic agents [10,11,12,13,14,15,16,17,18,19,20,21,22,23,24,25,26].

**Figure 2 life-14-00309-f002:**
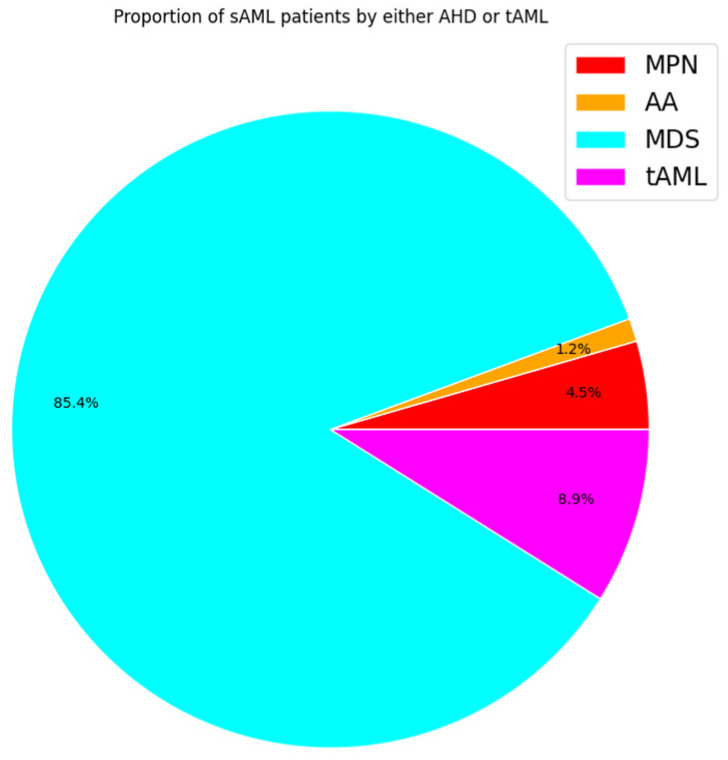
Pie chart displaying estimated proportions of sAML patients by history based on studies monitoring leukemic transformation. Each color represents the proportion of sAML patients by class. Percentages based on average incidence per 100,000 people.

**Figure 3 life-14-00309-f003:**
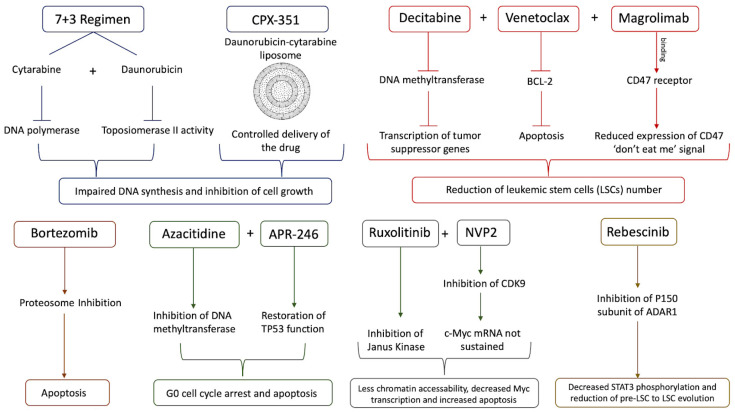
A comprehensive schematic representing several classes of drugs and their mode of action for sAML treatment.

**Figure 4 life-14-00309-f004:**
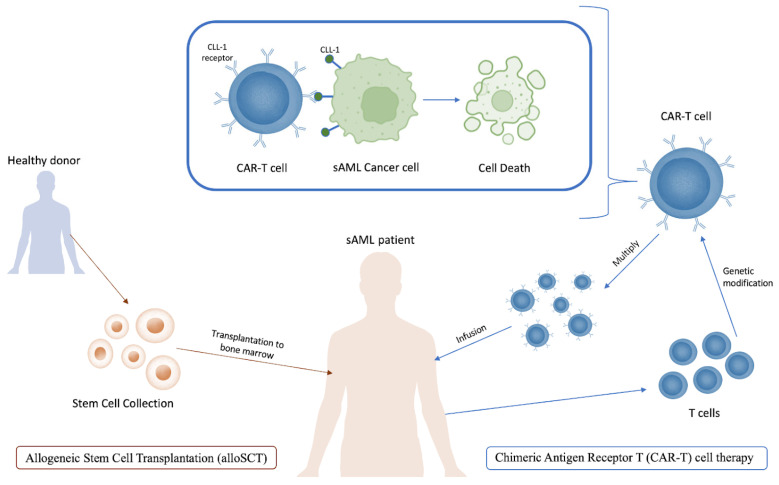
A diagram illustrating alloSCT (**left**) and CAR-T cell therapy (**right**) treatments for sAML.

**Table 1 life-14-00309-t001:** Hematological features associated with MPN, MDS, sAML and de novo AML.

Clinical Overview	Healthy	MDS	MPN	sAML	De Novo AML
Age (y)	18–65	53–98	18–92	21–77	18–59
White Blood Cells (109/L)	4–11	1.1–17.9	7.2–14.7	0.8–144.1	0.77–419.9
Platelets (109/L)	150–450	8–505	376–720	3–752	30–171
Hemoglobin (g/L)	120–175	47–149	109–173	34–143	2–1726
Clinical record		Free of treatment and transfusion	Both therapy free and treatment *	3 + 7 regimen Hypomethylation or palliative treatment	3 + 7 regimen
Reference	[62]	[63]	[64]	[65]	[65]

* anti-platelet drugs, cytoreductive therapy, JAK-2 inhibitors, immunomodulators, venipuncture, and supportive RBC transfusions.

## Data Availability

Not applicable.

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
