# Peer review of "Recent Advances towards the Understanding of Secondary Acute Myeloid Leukemia Progression"

_life, 2024, doi:10.3390/life14030309_

Round 1
Reviewer 1 Report
Comments and Suggestions for Authors
The authors report an interesting review on recent findings about some genetic and immunological abnormalities that characterize secondary AML and somehow differentiate them from "de novo" AML. Minor comments: 1) in line 242-244 a phrase report that mutated TP53, GATA and Kras have weak impact on AML progression and survival.. conversely, it is well known, and also reported in other points of the paper, that TP53 mutations hava a quite negative impact on AML prognosis. 2) in lines 252-253 it is reported that progression to AML occurs in 4% of essential thrombocythemia (ET) and 1% of policythemia vera (PV) patients, whereas it is usually believed (and also reported at the beginning of the paper) that AML evolution is more frequent in PV than in ET patients. 3) in lines 424-425 it is reported that allogeneic stem cell transplantation could be an alternative to classical "3 + 7" chemotherapy scheme...: it is not cleat what the authors mean, because allogeneic transplant usually gives the best results if performed after remission achievement, therefore when the patients have already received induction chemotherapy (either "3+7" or another scheme).
Author Response
1) in line 242-244 a phrase report that mutated TP53, GATA and Kras have weak impact on AML progression and survival.. conversely, it is well known, and also reported in other points of the paper, that TP53 mutations hava a quite negative impact on AML prognosis.
Our response: First of all we thank the reviewer for the valuable comments. We agree with the reviewer that TP53 mutations have a negative impact on AML and we have mentioned that in the first version of the manuscript in lines 276 and 277. However in this large study that comprises 2250 patients, the authors showed that TP53 mutations have a weak impact. To avoid any confusion we edited the sentence and removed the impact of TP53 shown in this study.
2) in lines 252-253 it is reported that progression to AML occurs in 4% of essential thrombocythemia (ET) and 1% of policythemia vera (PV) patients, whereas it is usually believed (and also reported at the beginning of the paper) that AML evolution is more frequent in PV than in ET patients.
Our response: We thank the reviewer for pointing out this discrepancy in the rates. We edited the sentence to: The rate of leukemic progression varies largely between the three subtypes of MPNs, with MF patients having the highest rate and ET the lowest .
3) in lines 424-425 it is reported that allogeneic stem cell transplantation could be an alternative to classical "3 + 7" chemotherapy scheme...: it is not cleat what the authors mean, because allogeneic transplant usually gives the best results if performed after remission achievement, therefore when the patients have already received induction chemotherapy (either "3+7" or another scheme).
Our response: We agree with the reviewer that the sentence was not clear enough and we edited it to: Nilsson et al. demonstrated the high potential of alloSCT when applied after chemotherapy.
We again thank the reviewer for carefully reading the review, which helped us a lot in improving its quality.
Reviewer 2 Report
Comments and Suggestions for Authors
Thanks for asking to review manuscript entitled: " Recent Advances Towards the Understanding of Secondary 2 Acute Myeloid Leukemia Progression" by Scott Auerbach , Beana Puka , Upendarrao Golla and Ilyas Chachoua.
AML is presenting a major challenge, with increase number of patients (with age), and lack of cure. The understanding of molecular mechanisms, and current treatments and protocols, truly help both clinicians in the treatment of each patient and researchers in the development of new drugs. The manuscript is well organized, and keeps a very good balance of detailed knowledge - but not too much of details, so the reader can easily follow the broader perspective.
It can be accepted and published pretty much at current form. I may suggest two possible improvements:
1. Table 1 is presented with simple discrete numbers. There is a range for each parameter. Authors may add additional references, and also present the normal range of healthy human at relevant age. The confidence interval is critical to realize how indicative is each parameter per patient. It is even better to highlight the advantage of personal record for the clinical assessment of each patient.
2. Tratemnts are rapidly evolving. Authors may consider referring to the recent ASH meeting publications for this-year recommendations. These will most likely be further updated next ASH...so authors may best suggest the reader to keep updating. Life is interesting this way.
Thanks again for this very good manuscript- I am sure that many will find it good to read.
Author Response
1. Table 1 is presented with simple discrete numbers. There is a range for each parameter. Authors may add additional references, and also present the normal range of healthy human at relevant age. The confidence interval is critical to realize how indicative is each parameter per patient. It is even better to highlight the advantage of personal record for the clinical assessment of each patient.
Our response: First of all we thank a lot the reviewer for the positive feedback and nice words. We edited the table according to the reviewer's request and we totally agree with him that presenting the table this way adds a lot of information.
2. Tratemnts are rapidly evolving. Authors may consider referring to the recent ASH meeting publications for this-year recommendations. These will most likely be further updated next ASH...so authors may best suggest the reader to keep updating. Life is interesting this way.
Our response: We thank a lot the reviewer for drawing our attention to the latest findings presented in ASH. We indeed found very interesting studies that address sAML treatment by developing novel strategies. We added two of these studies to the perspectives section.